# The Requirements for Setting Up a Dedicated Structure for Adolescents and Young Adults with Cancer—A Systematic Review

**DOI:** 10.3390/curroncol32020101

**Published:** 2025-02-11

**Authors:** Lukas Rudolf von Rohr, Nadja Battanta, Cornelia Vetter, Katrin Scheinemann, Maria Otth

**Affiliations:** 1Division of Oncology-Hematology, Children’s Hospital of Eastern Switzerland, 9006 St. Gallen, Switzerland; nadja.battanta@kispisg.ch (N.B.); cornelia.vetter@kispisg.ch (C.V.); katrin.scheinemann@kispisg.ch (K.S.); maria.otth@kispisg.ch (M.O.); 2Department of Oncology, Birmingham Children’s Hospital, Birmingham B4 6NW, UK; 3Faculty of Health Sciences and Medicine, University of Lucerne, 6002 Lucerne, Switzerland; 4Department of Oncology, University Children’s Hospital Zurich, 8008 Zurich, Switzerland

**Keywords:** AYA, adolescents, young adults, cancer care, model

## Abstract

Adolescents and young adults (AYAs), often defined as those aged 15–39 years, face unique challenges in oncology that are often unmet by conventional care models. This systematic review examines evidence on establishing dedicated AYA oncology units, focusing on logistical, infrastructural, and personnel-related recommendations. A PRISMA-guided search of PubMed (2000–2024) identified seven studies that emphasized early stakeholder involvement and collaboration between pediatric and adult oncology teams to ensure comprehensive care. Multidisciplinary teams (MDTs) of oncologists, nurses, and psychosocial support staff were highlighted as essential to address AYA patients’ diverse needs. Care models varied, with some advocating consultation-based services and others supporting dedicated units. Priorities included increasing clinical trial enrollment, fertility counseling, and creating environments attuned to AYA patients’ social and psychological needs. Key barriers included limited funding, institutional resistance, and inadequate pediatric/adult team collaboration. Despite progress, the lack of standardized guidelines and long-term data on AYA unit efficacy remains a challenge. Further research is required to develop outcome metrics, refine care models, and enhance survival and quality of life for AYA cancer patients.

## 1. Introduction

Adolescents and young adults (AYAs) have been recognized as a distinct population in oncology with specific but often unmet needs. The National Institutes of Health and the National Cancer Institute define the AYA population as individuals aged 15 to 39 years [1,2,3]. However, different age ranges may be used depending on local and national contexts, with upper age limits set at 24, 25, or 39 years [4]. Due to this broad age range, AYA patients present with a heterogeneous spectrum of tumor types, characterized by distinct biological features and malignancies commonly observed in both the pediatric and adult populations. Pediatric types of cancer include, for example, acute lymphoblastic leukemia (ALL) or rhabdomyosarcoma, whereas adult cancer types include, for example, breast and colorectal cancer [4,5,6,7]. Despite sharing a diagnosis, the biology may be different in the AYA population. For ALL, for example, favorable cytogenetic abnormalities in children, such as high hyperdiploidy or ETV6-RUNX1 are less frequent in the AYA population [8]. Conversely, AYA patients show less-favorable factors such as BCR-ABL1 or the intrachromosomal amplification of chromosome 21 (iAMP21) more frequently [9]. The same is true for adult types of cancer. In breast cancer, for example, women aged under 40 years are more often diagnosed with larger, poorly differentiated, and endocrine receptor-negative tumors and, further, have more frequent nodal involvement than women over 40 [9].

In addition to the broad range of tumor types, the prevalence and incidence of cancer is higher in the AYA population than in children. As demonstrated by Coccia et al., approximately 11,000 children were newly diagnosed with cancer in 2015, accounting for 0.65% of all cancer cases in the US, whereas around 70,000 AYAs were newly diagnosed, based on SEER data from 1995 to 2015, representing 4.72% of all cancer diagnoses [7,10]. For 2022, Hughes et al. estimated that, worldwide, 1,300,196 AYA patients were newly diagnosed with cancer, with 377,621 cancer-related deaths [11]. The age-standardized mortality was especially high in low-income countries [11].

Despite these peculiarities of the AYA population, international treatment protocols and clinical trials have predominantly focused on pediatric or adult patients, which have been crucial for improving survival and reducing treatment-related toxicity in these age groups. AYA patients are frequently under-represented and are less often enrolled in clinical trials [12,13,14,15,16,17]. In pediatric trials, for example, individuals over the age of 18 years are often excluded [12,13,14,15,16,17]. Furthermore, a substantial number of AYA patients are treated in general hospitals rather than specialized cancer centers that possess the concentrated expertise required for their treatment [18,19,20]. Although overall survival rates for AYA patients have improved in recent decades, progress for many tumor types has lagged behind other age groups. The lower inclusion rate in clinical trials, the high variety and uniqueness of different tumor types, and the fact that many AYA cancer patients are not referred to or treated in specialized oncological centers may partly explain this lower survival rate [7,10,21,22,23]. In addition to biological and medical aspects, AYAs are at a critical stage in their lives, where psychosocial pressure can be immense. They are facing many psychosocial challenges, such as working on their education, achieving financial independence, forming relationships, and in some cases managing family responsibilities. A cancer diagnosis can profoundly disrupt these developmental milestones, leading to a marked decline in quality of life [5,24,25,26]. Additionally, AYAs with cancer are in a more vulnerable state for psychological stress than their peers, and the risk for suffering from psychological problems is increased [27]. Also, compared to long-term survivors from childhood cancer they experience more post-traumatic stress disorder, anxiety, depression, and fatigue [27]. Even though AYA cancer patients are facing these immense challenges, there is still some limited recognition among healthcare providers about the unique psychosocial challenges and needs for this age group [4].

The establishment of dedicated AYA units has thus gained increasing importance, aiming to consider and better treat physical and psychosocial aspects and to improve the overall care experience for this population. Haines et al., for example, recently published a practical guidance to further develop and implement AYA units in the US [28].

Current models of care in pediatric and adult oncology settings, such as family-centered approaches in pediatric settings, are not tailored to meet the specific needs of AYA patients [14,29,30,31]. This group has been described as being in a “No Man’s Land”, receiving care that often fails to address their unique needs [32]. AYA units are intended to be specialized, designed to deliver age-appropriate care, comprehensive support, and resources that cover both the clinical and psychosocial needs of AYA patients. These units aim to create an environment that goes beyond pure cancer treatment to address the broader psychological and social challenges faced by AYA patients.

Despite the increasing adoption of AYA units, there remains a critical need for a systematic evaluation of their structure, implementation, and effectiveness. To address this gap, we conducted a systematic literature review to assess and evaluate the available evidence for recommendations to establish new AYA units dedicated to oncological care.

## 2. Materials and Methods

We conducted a systematic literature search in PubMed in December 2023, following PRISMA guidelines [33]. The search was restricted to studies published between 1 January 2000, and 1 January 2024. A search update was performed in November 2024 and included publications up to October 2024. The search strategy focused on three key concepts: “tumors/oncological diseases”, “adolescents and young adults”, and “set up/models of care/delivery of health” (Appendix A).

Inclusion criteria were defined using the PICO framework. The population of interest comprised AYA patients, defined as individuals aged 15–39 years at diagnosis and treated in dedicated AYA units for any cancer diagnosis. Studies that specifically addressed AYA patients but used alternative age ranges were also included. We did not specify and include comparators in the PICO framework as this is not a relevant factor for the questions of this systematic review, and we had already initially assumed that it would therefore not be mentioned in most of the eligible studies. The intervention was defined as the structure and set up of AYA units or programs, encompassing logistical, infrastructural, and personnel-related components. The primary outcomes were the age definitions used for AYA patients, as the age definition for this group is an often-discussed point in the oncology community and differs between different countries, and the recommendations for establishing new AYA units. Relevant parameters for the latter included logistical aspects (e.g., AYA unit located within pediatric or adult hospitals, primary medical lead), infrastructural elements (e.g., facilities such as recreation rooms), and personnel considerations (e.g., specialized training, availability of social workers). Based on the extracted data, we planned to analyze potential indicators of the medical, financial, or political benefits of establishing AYA units as secondary outcomes if possible.

Four authors (C.V., M.O., N.B., L.R.v.R.) independently screened all titles, abstracts, and full texts. Each publication was reviewed by two authors, with discrepancies resolved by a fifth reviewer (K.S.). Data were extracted from eligible studies into a standardized data sheet, including details such as first author, publication year, study design, patient characteristics, set up recommendations, and quality indicators, where available. Each study’s quality, relevance, and reliability were assessed by two authors. We used the Joanna Briggs Institute’s critical appraisal tools (https://jbi.global/critical-appraisal-tools, accessed on 8 July 2024), appropriate to each study type (e.g., appraisal tool for textual evidence narrative). Although the tool does not categorize studies by quality, we established a grading system: studies scoring five or six points (out of six) were classified as “Quality 1” (high quality), those with three or four points as “Quality 2” (medium quality), and those scoring one or two points as “Quality 3” (low quality).

Outcomes were planned to be described narratively based on the extracted data. This systematic review was registered with PROSPERO (https://www.crd.york.ac.uk/prospero/; CRD42024505963, accessed on 4 February 2024).

## 3. Results

The systematic literature search identified 2564 records, with an additional 92 records included from reference screening of the identified reviews. Following the title and abstract screening, 94 records were assessed for eligibility. Ultimately, 7 original articles were included in the final analysis, while 87 records were excluded—primarily due to outcomes other than recommendations for setting up an AYA unit (n = 56) or being review articles (n = 23) (Figure 1). All studies received six points in the quality assessment and were classified as Quality 1.

### 3.1. Study Characteristics, Age Definition, First Steps, Multidisciplinary Teams, and Models of Care

All the included studies were of a descriptive nature, mainly being narratives or expert opinions [34,35,36,37,38,39,40]. There were different age definitions of AYA patients used in the studies. The lower age cutoff ranged from 13 to 16 years and the upper limit from 24 to 39 years [34,35,36,37,38,39,40]. The reasons for choosing the age ranges were not mentioned (Table 1). The early involvement of political and medical stakeholders and leaders was identified as a critical early step in five studies (Table 1) [34,35,37,39,40]. Additionally, the concept of an AYA advocate or champion was highlighted in four studies [34,37,39,40]. These advocates/champions can, for example, conduct a multidisciplinary analysis of the current state of AYA care, initiate the planning of future services, advocate for resources, promote evidence-based clinical care, educate clinical personnel, and raise awareness (Table 1). The importance of having a clear financial situation from the very beginning was highlighted in three studies, also to prevent dissatisfaction in the involved personnel [35,36,40].

Collaboration between adult and pediatric oncology teams from the very beginning, already at the stage of the initial planning of the AYA unit, was universally recommended to ensure comprehensive care for AYA patients (Table 1) [34,35,36,37,38,39,40]. All studies stressed the importance of establishing a multidisciplinary team (MDT) to address the complex and diverse needs of AYA oncology patients [34,35,36,37,38,39,40]. Across studies, key MDT members included lead clinicians, nurses, pediatric and adult oncologists, psychosocial support teams, and allied health professionals, such as dietitians, social workers, and physiotherapists (Table 1) [34,35,36,37,38,39,40].

While no specific model of care was universally recommended, the studies emphasized the need to adapt and evolve care models based on local circumstances, reflecting the variability in available resources and patient demographics (Table 1) [34,35,36,37,38,39,40]. Haines et al., for example, described a “consultation-based” model allowing AYA-focused services to be available across disease groups and settings [37]. The service spanned inpatient and outpatient settings in pediatric and adult oncology. Osborn et al. describe the situation of five centers in eight Australian states where the models differ depending on the jurisdiction: multiple hospital-based lead sites, a collaborative network partner model, a single statewide service (mobile team) working across adult and pediatric sectors and across regions, and a single lead site with a statewide responsibility [38].

### 3.2. Clinical Trial Inclusion, Logistical Recommendations, and Further Aspects

The inclusion of AYA patients into clinical trials emerged as a central theme, with six studies stressing the importance of increasing participation due to the historical under-representation of AYAs in clinical research (Table 2) [34,35,36,37,38,39]. Carr et al. proposed an “informed opt-out” consent model to enhance trial enrolment, while both Magni et al. and Haines et al. highlighted the need for trial protocols specifically tailored to AYA patients [34,36,37]. Five studies advocated for the development of AYA communities or networks to facilitate the creation of treatment guidelines and educational programs [34,36,37,39,40]. Creating an AYA-friendly environment was mentioned in six studies [34,35,36,37,38,40]. This includes aspects of clinical care (e.g., ward round time, visiting hours) and social aspects (physical and digital interaction among AYA patients, rooms/meeting areas designed and equipped for AYAs) (Table 2). Windebank et al. even highlighted that AYA patients should be included in the design and equipment of AYA units, but also in formulating ward rules [35]. Finally, four studies mentioned or recommended the development of quality indicators to assess the effectiveness of AYA care models and initiatives (Table 2) [36,37,39,40].

Fertility counseling was specifically mentioned in six studies, with Haines et al., for example, recommending the inclusion of a dedicated AYA fertility specialist [35,36,37,38,39,40]. Additionally, four studies emphasized the importance of structured survivorship care [35,36,37,39]. Scott et al. specifically mentioned the inclusion of AYA survivors, who were diagnosed as young children with an oncological disease, as these patients are often forgotten in survivorship care [39].

### 3.3. Barriers and Facilitators in Setting Up AYA Units

The primary barriers to establishing AYA oncology units were identified as insufficient collaboration between pediatric and adult oncology teams, a lack of awareness, challenges in changing existing institutional medical and political cultures, and a lack of funding (Table 3) [34,35,36,37,38,39,40].

Facilitators included the establishment of national AYA networks and communities, having financial and political support, an analysis of the current situation, the engagement of key stakeholders including healthcare professionals, allied health staff, and also patient and caregiver advocacy [34,35,36,37,38,39,40]. The development of AYA oncology as a recognized specialty was suggested as a long-term solution to some of the barriers to care [36].

## 4. Discussion

The results of this systematic review reveal the diversity of approaches to establishing AYA oncology units while identifying common essential components for successful implementation.

A key finding across all the studies was the early involvement of relevant stakeholders in the planning process, which was universally recognized as crucial to the successful establishment of AYA units. Through this approach, members from the pediatric and adult oncology team feel a sense of belonging right from the start, can actively contribute, and do not feel left out. Having an AYA champion from the start, as mentioned by several included studies, might facilitate these efforts [34,37,39,40,41,42]. Several studies highlighted the significant barriers posed by the lack of collaboration between pediatric and adult oncology teams and the challenge of shifting established institutional practices. Reed et al. even described it as a ‘turf war’ between medical and pediatric oncologists, arguing who is better suited to treat AYA patients [42]. Given the broad spectrum of tumor types in AYA patients, which are normally managed by either pediatric or adult oncology specialists, strong collaboration between these teams is essential to ensure optimal care for AYA patients. For example, studies have shown that AYA patients with acute lymphoblastic leukemia or sarcomas, which are more often seen in pediatric patients, achieve better survival outcomes when treated with pediatric protocols [23,43,44,45,46]. Furthermore, a well-known referral and transition pathway will facilitate the collaboration between the different pediatric and adult specialties and will help to increase the numbers of AYA patients treated in a specialized ward [37,39,41].

Another important factor in setting up an AYA unit is strengthening the AYA community at the local, national, and international levels. This strategy can increase acceptance and support for these efforts from both political and medical sectors. Moreover, it may lead to improved educational opportunities for the medical team and the development of more comprehensive guidelines, thus resulting in better care. Patterson et al. concluded that established national systems and coordination seem to lead to higher patient satisfaction, including age-appropriate information and support services for AYA patients, and specialist services [47]. Over the past few decades, significant progress has been made and European collaboration groups created, highlighted in an official position paper on AYA treatment in 2021 by both the adult European Society for Medical Oncology (ESMO) and the pediatric International Society of Pediatric Oncology (SIOP Europe) [48].

An interesting and consistent finding across the studies was the variation in age definitions for AYA patients, reflecting the long and ongoing debate over the most appropriate age range for this population. A consensus on a standardized age range could help to guide and standardize clinical care and research. The 2021 ESMO/SIOPE position paper on AYA care proposed an age range of 15–39 years to harmonize the internationally varying definitions [48].

However, the disadvantage of a broader age range implies different psychosocial needs and cancer biologies. Psychological needs differ widely between cancer patients aged 20 years and 39 years; for example, in terms of maturity, partnership and sexual life, and worries about the future. The same is true for social aspects such as education and career or economic independence. Regarding biological aspects, epithelial cancers are more commonly seen in those aged 25 years and older [49].

The recommended settings and models of care for AYA units also varied significantly. Some studies advocated for consultation-based models integrated within both pediatric and adult oncology services, while others suggested the establishment of dedicated lead sites serving regional or national populations [37,38]. Both Osborn et al. and Haines et al. emphasized that the appropriate model of care should be determined by local circumstances and the existing structures and resources available. Currently, there is no evidence to indicate which model of care is most favorable for AYA patients [37,38].

A consistent theme across all the studies was the importance of establishing a multidisciplinary team to address the complex needs of AYA patients. Especially, the need for psychosocial support and appropriate screening is repeatedly mentioned because the psychological burden of these diseases is high [27,28,34,35,36,37,38,39,40,41,42]. Additionally, creating age-appropriate care environments that foster social interaction and promote psychosocial well-being was emphasized. The focus on flexible designs and dedicated recreational areas reflects an understanding that AYA patients face not only medical challenges but also significant quality-of-life concerns during and after treatment [5,24,25,26]. This was, for example, also seen for AYA patients with chronic diseases. The psychological significance of AYAs with chronic illnesses being surrounded by peers of a similar age and having a role model within their age group has been demonstrated [50].

Another critical point discussed was the under-representation of AYA patients in clinical trials. Several studies emphasized the need to prioritize AYA trial inclusion when designing new AYA units. The “informed opt-out” consent model proposed by Carr et al., in which patients are automatically enrolled in trials unless they actively decline, could help increase participation. This approach could mitigate the current under-representation of AYA patients in oncology trials, which has contributed to stagnation in survival outcomes compared to other age groups. Addressing the limited number of trials open to AYA patients, often due to restrictive age criteria, is another critical step [12,13,14,15,16,17]. To tackle this problem, the Working Group on Fostering Age-Inclusive Research (FAIR) was launched in 2017 by the ACCELERATE Forum (https://www.accelerate-platform.org/fair-trials, accessed on 4 October 2024). The FAIR aims to raise awareness about the problem of the upper age limit of 18 years in pediatric cancer trials and to promote change. This age limit is arbitrary and without medical justification. A similar effort is also undergoing in the US with the “Children’s Oncology Group Adolescent and Young Adult Responsible Investigator Network” [51].

Fertility counseling was consistently identified as an important component of AYA care. The potential long-term impact of not offering fertility preservation is significant for this patient group, and multiple studies emphasized the importance of providing fertility counseling to all at-risk patients, which should be standard of care [5,52,53].

The limitations of this systematic review are linked to the data provided in the studies. Only seven studies were included as reviews could not be included for methodological reasons. The design, the included population, and the outcome of these studies were heterogenous. They were mainly retrospective analysis/narratives and expert opinions, which were conducted in a descriptive form, and no prospective data were collected. The strengths include the comprehensive approach by screening titles, abstracts, and full texts by two independent reviewers, and the detailed quality assessment of the included studies, ensuring reliability and relevance.

Despite these findings, there are still significant gaps in evidence-based guidelines for AYA oncology care. Only seven studies met the inclusion criteria for this systematic review, indicating a scarcity of data in this field. While the included studies provide valuable recommendations and experience for the structure and organization of AYA units, data on the long-term outcomes and effectiveness of these programs remain limited. Future research should focus on developing standardized metrics to evaluate the impact of AYA care models on patient survival, quality of life, and psychosocial outcomes. In a workshop carried out in 2011 by the Canadian National Task Force on Adolescent and Young Adult Oncology, supported by the Canadian Partnership Against Cancer and the C17 network, relevant categories of outcomes for AYA with cancer and respective metrics to assess them were defined [54]. The defined categories range from epidemiology to screening and prevention, access and place of care, psychosocial health and quality of life, and survivorship, but also economic aspects. Economic metrics to assess the cost/benefit ratio of care are crucial not only to improve the care for AYA patients but also to justify the financial support and funding needed to establish dedicated AYA structures [54]. Ferrari et al. evaluated eight different metrics in a single center in Italy [55]. They considered these metrics important so that AYA projects are accepted as standard of care by medical and political stakeholders [55]. The prospective, longitudinal, observational BRIGHTLIGHT cohort was launched in 2012 in the UK and prospectively collected data to evaluate the benefits of AYA services in the UK [56]. Aspects considered in this cohort included quality of life, satisfaction with care, clinical processes and outcomes, patients’ experience of cancer care, social and educational milestones, and the costs of care. Today, BRIGHTLIGHT has expanded beyond this initial cohort study and includes a broad range of studies and projects related to cancer care in the AYA population [56]. Charities can be additional sources for information about care structures and the AYA patients’ perspective. Teenage Cancer Trust in the UK, for example, published in 2012 a blueprint of care for teenagers and young adults with cancer [57]. Additional charities and organizations are “Young lives vs. Cancer”, “CanTeen”, and “Teen Cancer America”, some of which also provide information about desired care structures. A good overview about these structures is given by Ferrari et al., emphasizing the importance of these organizations [4].

## 5. Conclusions

In conclusion, while notable progress has been made in understanding the unique needs of AYA oncology patients, much work remains in formalizing the structure, funding, and collaboration required for the successful establishment of AYA oncology units. Further research is essential to evaluate the long-term outcomes and effectiveness of these programs.

## Figures and Tables

**Figure 1 curroncol-32-00101-f001:**
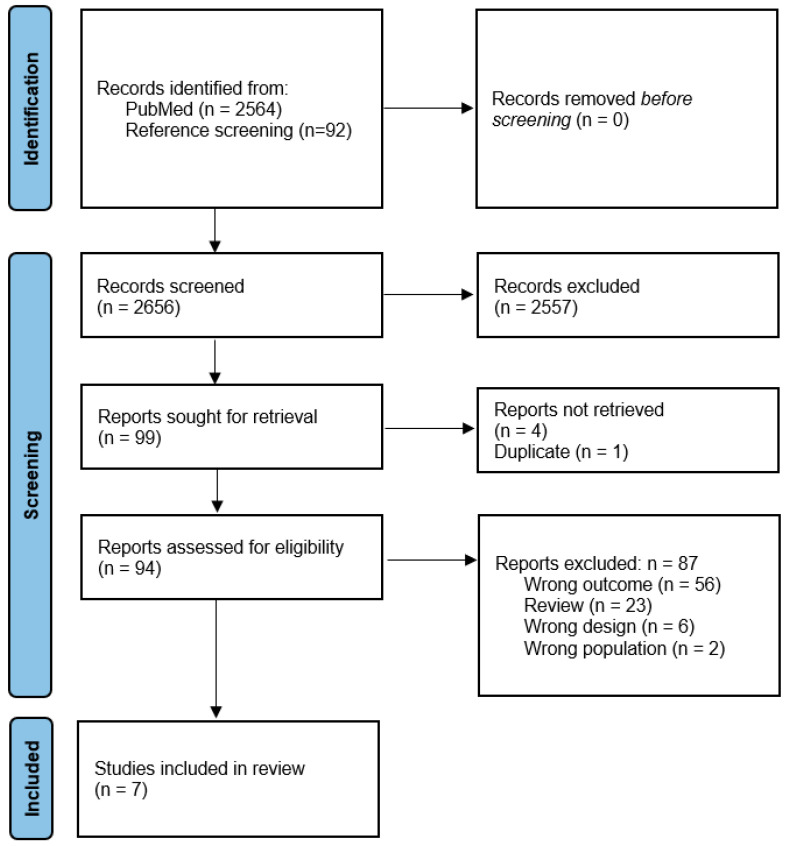
Flow diagram of included studies for review of AYA unit set ups.

**Table 1 curroncol-32-00101-t001:** Characteristics of included studies, information about initial steps, and composition of multidisciplinary teams.

Author (Year of Publication)	Study Design, Included Centers	Quality	Age Definition	Recommended Initial Steps	Recommended MDT Members/Specialties Involved	Recommended/Described Setting/Model of Care
Carr et al. (2013) [34]	Descriptive, 25 TYA cancer centers in UK	High quality	16–24 years	Involve important stakeholders (political and medical)	Lead clinician and nurse, adult/pediatric oncologists, psychosocial team, social worker, youth worker, dietician	TYA cancer centers UK
Windebank et al. (2004) [35]	Descriptive, centers in Leeds and Newcastle, UK	High quality	No age range defined. In 14 centers, range was 10–25 years	- Multidisciplinary planning group incl. relevant disciplines - Involve political and medical stakeholders - Clear AYA definition and expected patient numbers - Clear financial situation - Decide as team on treatment guidelines	Lead clinician and nurse, adult/pediatric oncologists, psychosocial team, social worker, dietician, pharmacist, activity coordinator, community liaison team, managers	Adapt to local setting, evolve model of care with help of adult and pediatric team
Magni et al. (2016) [36]	Descriptive, Istituto Nazionale dei Tumori, Milano	High quality	15–29 years	- Establish collaboration between adult/pediatric team - Clear financial support	Collaboration of adult and pediatric oncologists, nurses, psychosocial team, social worker, physiotherapist, radiotherapists, surgeons, fertility and sexuality consultant, teachers, educators, spiritual assistant	Youth project of Istituto Nazionale Tumori in Milano; AYA-specific model of care needed, in collaboration with adult and pediatric oncologists
Haines et al. (2023) [37]	Descriptive, Lineberger Comprehensive Cancer Center, University of Carolina	High quality	13–39 years	- AYA advocate/champion - Analyze current situation (e.g., resources? gaps? possible key person?) - Involve institutional leaders early - Find common ground between different disciplines, establish common guidelines - Start small - Patient advisory board	Build up an AYA-specific team according to local needs and possibilitiesPediatric and adult oncology, psychosocial support, AYA nurse practitioner, AYA social worker, medical oncology liaison, palliative care physician	Model of care and setting according to local circumstances. Consultation-based model: AYA-specific service available across age and disease groups (in- and outpatient, pediatric and medical oncology)
Osborn et al. (2013) [38]	Descriptive, five centers in eight Australian states (Victoria, South Wales, South Australia, Queensland, Western Australia)	High quality	15–25 years	- Establish lead AYA cancer care site(s) across country - Access to support services and clinical trials - Comprehensive assessment at diagnosis - Coordinated care to empower AYA decision making - Expert multidisciplinary team skilled in AYA cancer care; discuss every patient at MDT meeting; key person for every patient assigned	Pediatric and adult oncologists, program manager, social worker, nurses, ward staff, psychologist, educational and vocational officer, exercise physiologist, researcher, nurse coordinator, medial fellow, music therapist	Models differ depending on jurisdiction Multiple hospital-based lead sites Collaborative network partner model: lead sites work with integrated cancer services across regions and offer secondary consultation services to other hospitals Single statewide service working across adult and pediatric sectors and across regions (mobile team)
Scott et al. (2024) [39]	Descriptive, Department of Pediatric Hematology-Oncology, CancerCare Manitoba, Winnipeg	High quality	15–39 years	- Analyze current situation, prioritize key gaps with realistic goals (e.g., onco-fertility, psychosocial care, clinical trial inclusion) - Scan of existing resources - Strategic planning document - Stakeholder involvement	Oncologists, AYA champions, sexuality and fertility counselor, palliative care, survivorship care, clinical nurse specialist, dietitian, research coordinator, social service, occupational therapist and physiotherapist, psychological support	Provincially mandated agency providing clinical services for patients in different hospitals
Smith et al. (2024) [40]	Descriptive, Department of Pediatric Hematology-Oncology, Princess Margaret Cancer Centre, University of Toronto	High quality	15–39 years	- Assessment of current situation (resources? gaps?) - Involve hospital key persons/stakeholders and organizations - Lobby for funding - Focus on initial priorities - Develop referral processes internally and externally	Program leader/AYA champion, medical and pediatric oncologist, clinical nurse specialist, social worker, psychiatrist, researcher, program coordinator, school/work transition counselor, rehabilitation medicine, radiation oncologist, palliative care specialist, spiritual care	Large central cancer center with partnership with regional cancer centers (hub and spoke model). Central cancer centre provides core staff/resources and also training to regional hospital

**Table 2 curroncol-32-00101-t002:** Summary of recommended logistics/assessment/education, clinical trial inclusion, fertility counseling, and additional aspects.

Author	Recommended Logistics/Environment	Recommended Assessment/Quality Indicators	Education and Network	Clinical Trial Inclusion	Fertility Counseling	Others
Carr et al. (2013) [34]	- Hospital units capable of caring for critically ill patients - Environment to encourage social interaction among AYAs (physically, digitally) - Environment where TYA patients have some control - Personal space designed for young people with the adequate equipment and privacy	TYA cancer registry	- Create national AYA community - Peer-reviewed guidelines - Educational courses - Care of AYAs as own specialty	- Increase inclusion rate (‘informed opt-out consent’) - Separate clinical study group for TYAs	Not mentioned	Survivorship not mentioned
Windeb-ank et al. (2004) [35]	- Flexible design (e.g., bedroom) dependent on local setting and patient numbers - AYA-friendly hospital daily routine (ward round time, visiting hours, etc.) and meeting area - Private area/lounge (e.g., video, music, games) - AYAs help for design and ward rules - Space for families	Not stated	Not stated	Must be improved	Must be improved	UKCCLG for survivorship care. Decide on protocol regimen (adult? pediatric? AYA oncologist?)
Magni et al. (2016) [36]	- Dedicated spaces and projects - Multifunctional room for socializing (tv, music, computer, books)	Measure of efficacy by indirect parameters (e.g., trial inclusion, fertility counseling, psychosocial support, patient satisfaction)	AYA specialist as goal, strengthening of AYA networks	Availability of clinical trial protocols for all tumor types in AYA patients	Offered to all patients at risk	Survivorship care according to risk profile Support projects (fashion, writing classes, sports projects, etc.)
Haines et al. (2023) [37]	Not stated	Formulation of metrics to evaluate program (no example stated)	Annual symposium for MDT members and researchers to educate and increase engagement	Increase trial inclusion	AYA fertility specialist, fertility preservation group	Survivorship clinic directly after end of treatment
Osborn et al. (2013) [38]	- Specific AYA cancer in- and outpatient rooms - Teenage lounge - Decoration by former patients - Portable equipment (e.g., film, gaming console)	Not stated	Not stated	High priority within contract between CanTeen and the Clinical Oncology Society of Australia	High priority for fertility preservation	Survivorship not mentioned
Scott et al. (2024) [39]	Not stated	Evaluation of program with different metrics (e.g., clinical trial inclusion, psychosocial support, onco-fertility referral)	- Collaborate with existing regional/national/international structures - Consultation and education of other specialist groups to emphasize importance of AYA program and find new AYA champions	- AYA research coordinator - AYA research platform	High priority for onco-fertility and established referral pathway	Survivorship care for all AYA patients (including those diagnosed as children)
Smith et al. (2024) [40]	- Facilitating peer connections through virtual and non-virtual events - No recommendations about set up in hospital	- Patient satisfaction - Patient referral numbers - Patient attendance numbers at social events - Fertility preservation rates	- Local education programs with clinical nurse specialists - Further education for regional hospitals through central hospital	Not stated	Referral pathway for onco-fertility counseling for all patients	Establish rehabilitation program and palliative care

**Table 3 curroncol-32-00101-t003:** Summary of barriers and facilitators in setting up AYA structure.

Authors	Barriers/Challenges	Facilitators
Carr et al. (2013) [34]	- Lack of scientific data - Lack of funding - Insufficient communication between different teams (e.g.: tumor group MDT and TYA MDT)	- AYA network - Building national communities/structures to strengthen AYA efforts (make the problem public) - Local ‘ambassador/champion’ to support TYA-ward
Windebank et al. (2004) [35]	- Insufficient collaboration between specialties	- Include important stakeholders early on to find common path in treatment and decide on which group provides which service (e.g., CVD implantation in a 20-year-old: pediatric surgeon? adult surgeon? interventional radiologists?)
Magni et al. (2016) [36]	- Co-operation between pediatric and adult oncologists is hard to achieve due to different working methods, different priorities, and different classification, staging, and grading systems	- AYA oncologist as new specialty - MDT with collaboration between adult and pediatric oncologists
Haines et al. (2023) [37]	- Changing existing culture - Funding - Building new team	- Foundations and philanthropic support - Start small - Strategic plan before presentation to leadership/other members - Leveraging patient and caregiver advocacy - Meet institutional leaders and educate them - AYA champions
Osborn et al. (2013) [38]	- Demographic situation of Australia: long distances hinder centralization; individual physicians see few AYA patients only - Collaboration between adult and pediatric oncology - Developing national model; different needs of different jurisdictions - Not enough evidence for patient group - Not enough available clinical trials for patient group	- Support by government (Cancer Australia), including funding - Engagement of key stakeholders (pediatric and adult oncologists, allied health staff), building relationships - Understanding of referral pathways - AYA champions - Educational framework for awareness and expertise
Scott et al. (2024) [39]	- Not stated	- Strategic planning document helps bring attention to program and gather support from all stakeholders - Connecting key individuals from different departments early - Collaboration between adult and pediatric oncology - Narrow focus of program initially might lead to expansion later - Develop new referral pathway and show/explain it to different departments/referring institutions - Engage AYA to recognize gaps and to find future goals
Smith et al. (2024) [40]	- Lack of awareness of specific needs of AYA patients - Securing sustainable funding is critical - Creating pathways and connections in regional areas can be difficult	- Early involvement of stakeholders, especially government and policymakers, is imperative

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
