# Peer review of "The Requirements for Setting Up a Dedicated Structure for Adolescents and Young Adults with Cancer—A Systematic Review"

_curroncol, 2025, doi:10.3390/curroncol32020101_

Round 1
Reviewer 1 Report
Comments and Suggestions for Authors
This article summarizes the unique challenges faced by adolescents and young adults with cancer, and the attempts to date to offer comprehensive medical and psychosocial oncology care to this population. The methods are rigorous, and the results are presented clearly. The gap in knowledge is well-characterized, and there is high relevance to the field.
The introduction provides a good overall summary of the issues in AYA care. Some more recent references could be included, such as Hughes et al. 2024 doi: 10.1016/S1470-2045(24)00523-0, Fidler et al. 2019 doi: 10.1002/pbc.27668, Close et al. 2019 doi: 10.3322/caac.21585, Ferrari and Barr. 2017 doi: 10.1002/pbc.26528. The authors may consider replacing the word “somatic” (lines 65 and 72) with “medical”, “physical” or “biological”. While somatic does mean “having to do with the body”, it may lead some readers to think about somatization, as in the expression of psychological or emotional factors as physical symptoms.
The methods were well-described, PRISMA guidelines were followed, the review was registered with PROSPERO, a flow diagram was presented, and the review process including the use of an appraisal tool was rigorous. When describing the PICO framework, the authors could consider specifying that there was no comparison group in this study. Also, the authors could clarify why age definitions used for AYA patients was considered an outcome.
The results were presented very clearly. Figures and tables were appropriate and easy to understand and interpret.
The discussion contextualizes the results well. However, while this review was limited by the available data, which the authors acknowledge clearly, the discussion could go deeper. The authors could consider including more about metrics (see, for example, Greenberg et al. 2011, doi: 10.1002/cncr.26040). The discussion could also include references to relevant, recent articles that did not meet the strict inclusion criteria but contribute to a more fulsome picture of the state of AYA programs and units around the world, such as:
Patterson et al 2023 doi: 10.1089/jayao.2022.0182.
Haines et al 2023 doi: 10.1089/jayao.2023.0110.
Surujballi et al 2022 doi: 10.3390/curroncol29060322
Haines et al 2022 doi: 10.1200/OP.22.00063.
Fardell et al 2018 doi: 10.1089/jayao.2017.0100.
Reed et al 2017 doi: 10.1089/jayao.2017.0051.
Thank you for your excellent work!
Reviewer 2 Report
Comments and Suggestions for Authors
Thankyou for the opportunity to review this systematic review of the requirements for setting up a details structure for adolescents and young adults with cancer. It is nicely written.
In doing this piece of work the authors have followed standard systematic review methodology. One drawback for this study is that this meant only 7 studies were considered as having relevant, good quality evidence and were included in the review. Whilst the methodology is sound, to me this approach is not inclusive enough to capture the wealth of other data that exist to inform the topic of how to set up AYA services, particularly data that come from research by UK charities such as the Teenage Cancer Trust (see their Blueprint of Care for TYA Cancer) and Young Lives vs Cancer and the voice of patients and patient advocates which is really strong for AYA care. I wonder whether a trawl of "softer" data to add to the results of the systematic review might make an even more balanced read in terms of what is needed to set up an AYA unit, although I think that the systematic review has picked up the main points. The need for psychological support is massive and this doesn't come out sufficiently in the paper. Neither is there any discussion about the need for good pathways for transition from paediatric to TYA and TYA to adult services. Finally, at least in some places there has been an effort to evaluate the efficacy of AYA services - for example the UK BRIGHTLIGHT data - and it would be helpful to mention such studies in the discussion.
Other specific minor points
Line 36, comma after "both" is not needed
Lines50-51 - please be clearer here about time frames - when are these data on numbers of diagnosis from and is this eg per year or in one specific year?
Round 2
Reviewer 1 Report
Comments and Suggestions for Authors
Thank you for your diligent efforts to address the suggestions. Very well done!
Reviewer 2 Report
Comments and Suggestions for Authors
Thankyou for the opportunity to review the revised manuscript. I am happy with the changes made. Well done for this nice piece of work